# Identification and Functional Characterization of a Putative Alternative Oxidase (Aox) in *Sporisorium reilianum* f. sp. *zeae*

**DOI:** 10.3390/jof8020148

**Published:** 2022-01-31

**Authors:** Hector Mendoza, Caroline D. Culver, Emma A. Lamb, Luke A. Schroeder, Sunita Khanal, Christian Müller, Jan Schirawski, Michael H. Perlin

**Affiliations:** 1Department of Biology, Program on Disease Evolution, University of Louisville, Louisville, KY 40292, USA; hector.mendoza@louisville.edu (H.M.); caroline.culver@louisville.edu (C.D.C.); emma.lamb.1@louisville.edu (E.A.L.); luke.schroeder@louisville.edu (L.A.S.); 2Division of Cardiovascular Medicine, School of Medicine, University of Maryland, Baltimore, MD 21201, USA; skhanal@som.umaryland.edu; 3Matthias Schleiden Institute, Friedrich-Schiller University, 07737 Jena, Germany; christian-mueller@uni-jena.de (C.M.); jan.schirawski@uni-jena.de (J.S.)

**Keywords:** alternative oxidase, basidiomycete, respiration, AMA, SHAM, teliospore

## Abstract

The mitochondrial electron transport chain consists of the classical protein complexes (I–IV) that facilitate the flow of electrons and coupled oxidative phosphorylation to produce metabolic energy. The canonical route of electron transport may diverge by the presence of alternative components to the electron transport chain. The following study comprises the bioinformatic identification and functional characterization of a putative alternative oxidase in the smut fungus *Sporisorium reilianum* f. sp. *zeae*. This alternative respiratory component has been previously identified in other eukaryotes and is essential for alternative respiration as a response to environmental and chemical stressors, as well as for developmental transitionaoxs during the life cycle of an organism. A growth inhibition assay, using specific mitochondrial inhibitors, functionally confirmed the presence of an antimycin-resistant/salicylhydroxamic acid (SHAM)-sensitive alternative oxidase in the respirasome of *S. reilianum*. Gene disruption experiments revealed that this enzyme is involved in the pathogenic stage of the fungus, with its absence effectively reducing overall disease incidence in infected maize plants. Furthermore, gene expression analysis revealed that alternative oxidase plays a prominent role in the teliospore developmental stage, in agreement with favoring alternative respiration during quiescent stages of an organism’s life cycle.

## 1. Introduction

*Sporisorium reilianum* is a phytopathogenic basidiomycete with two *formae*
**s***peciales*, able to infect maize (*S. reilianum* f. sp. *zeae*, SRZ) and sorghum (*S. reilianum* f. sp. *reilianum*, SRS). Much like its better known relative, *Ustilago maydis*, SRZ displays a dimorphic lifestyle, existing as yeast-like haploid sporidia that reproduce asexually by budding [1]. Sexual reproduction can take place and is dependent on a tetrapolar mating-type system that begins with a pheromone-pheromone receptor interaction and allows the fungus to enter its pathogenic stage, first as dikaryotic hyphae, and eventually, as diploid teliospores that can germinate under the right conditions and restart the life cycle [2]. Unlike *U. maydis*, however, SRZ causes systemic infection on the plant host, emerging as teliospore sori on both male (tassel) and female (ear) inflorescences and causes additional developmental abnormalities, such as phyllody and irregular branching architecture [3,4].

The bioenergetics and intermediary metabolism of fungal systems are poorly understood, despite the relevance of mitochondria and the associated electron transport chain (ETC) as the main source of metabolic currency. Like its mammalian counterpart, the fungal “powerhouse” is comprised of the four classical protein complexes (NADH ubiquinone oxidoreductase or complex I, succinate dehydrogenase or complex II, cytochrome bc_1_ or complex III and cytochrome c oxidase or complex IV) that can function as proton pumps (with the exception of complex II) to help maintain the electrochemical gradient that drives ATP synthesis via oxidative phosphorylation. Pumping of protons is facilitated by the fixed movement of electrons through the protein complexes, from NADH as the initial donor to molecular oxygen as the final acceptor.

Alternative respiratory pathways recently received special attention, in which additional components diverge electron movement from the classical cytochrome c pathway. For instance, plants were known to deviate from the norm in response to specific metabolic needs, as is the case of the peanut plant, *Arachis hypogea*, whose embryos are deficient in cytochrome c, are insensitive to potassium cyanide (KCN) and are able to use NADH and succinate as substrates for uncoupled respiration, with significantly reduced ATP production [5]. Another notable alternative component of the ETC was discovered early in a variety of yeasts, which are among the only organisms devoid of complex I [6]. Instead, yeasts have internal and external rotenone-insensitive NADH dehydrogenases that utilize matrix NADH as substrate. 

The alternative respiratory component first identified in the peanut plant is a quinol oxidase, also known as alternative oxidase (Aox), which functions like complex IV and catalyzes the reduction of oxygen into water but does not function as a proton pump [7]. In this manner, electron flow is limited to complexes I and II, with complex I being the sole contributor to the electrochemical gradient in the mitochondrial intermembrane space. In plants, deviation from the cytochrome c pathway has been associated with thermotolerance [8,9], chemical [10,11] and oxidative [12,13] stresses, and immunity [14,15]. Aox was also identified in parasites like *Trypanosoma brucei* (Trypanosome alternative oxidase, TAO) [16], in which it is essential for the maintenance of the mammalian bloodstream forms of the pathogen, and may prove a crucial target in the development of novel drug therapies for the sleeping sickness caused by this parasite [17,18]. Aox was also identified in *Cryptosporidium parvum* (CpAox), the causative agent of cryptosporidiosis, that lacks membrane-bound organelles and is solely dependent on Aox-mediated respiration limited to mitochondrion-like compartments (mitosomes) [19,20].

In the case of fungi, Aox has been identified in most annotated genomes, including a variety of agriculturally relevant phytopathogens like *Botrytis cinerea* [21], *Magnaporthe grisea* [22], *Mycosphaerella graminicola* [23] and several *Aspergillus* species [24,25,26], making it an excellent fungicidal target for crop management approaches. Aox was also suggested to be involved in developmental regulation, as is the case for the fungus *Blastocladiella emersonii*, in which respiratory capacity is higher in highly motile zoospores, compared to vegetative cells that favor Aox-mediated respiration due to their lower metabolic demands [27].

The aim of this study was the identification and functional characterization of a putative Aox in SRZ, based on previous studies involving the related species *U. maydis* [28,29,30], which found that Aox is dispensable for overall welfare of the fungus but crucial during respiratory stress. SRZ differs from *U. maydis* in the way it infects maize, with symptoms appearing during the late stages of maturity of the plant. This peculiarity makes SRZ an excellent model to determine if Aox is involved in the long-term survival of the fungus in the plant tissue. Additionally, expression analysis of distinct developmental stages of the fungus indicated increased *aox* transcript levels in teliospores, corresponding with the use of a less energy-providing respiratory pathway during quiescent stages of the life cycle. 

## 2. Materials and Methods

### 2.1. Strains and Growth Conditions

The SRZ strains used and generated in this study are listed in Appendix A. Haploid strains were grown in potato dextrose (PD) broth on a rotary shaker at 200 rpm or on solid PD agar at 28 °C. Transgenic strains were selected by supplementing the PD medium with hygromycin or carboxin at a final concentration of 100 µg/mL or 5 µg/mL, respectively. For cloning purposes and plasmid maintenance, the transformed *Escherichia coli* DH5α strain (Thermo Fisher Scientific, Waltham, MA, USA) was grown in Lysogeny Broth (LB) on a rotary shaker at 200 rpm or on solid LB agar at 37 °C. Plasmid selection was achieved by supplementing the LB medium with ampicillin at a final concentration of 200 µg/mL or gentamicin at a final concentration of 10 µg/mL.

### 2.2. Bioinformatics

The amino acid sequence of the *U. maydis* Aox (Aox1, NCBI Accession No. XP_011389130.1) was used in a BLASTp search against the genome of SRZ2 (GCA_000230245.1). The resulting amino acid sequence was then used for Multiple Sequence Comparison by Log-Expectation (MUSCLE) in SnapGene 5.3.2 (Insightful Science, LLC, San Diego, CA, USA). For comparison, the amino acid sequences of Aox in other species were included: *Sporisorium scitamineum* Aox (CDU22616.1), *Sporisorium graminicola* Aox (XP_029737873.1), *Ustilago bromivora* Aox (SAM82122.1), *U. maydis* Aox1 (XP_011389130.1) and *Pseudozyma hubeiensis* Aox (XP_012186999.1).

### 2.3. Molecular Techniques

DNA isolation from SRZ and transformation protocols were carried out as previously described for *U. maydis* [31]. Primers used for the generation of the individual fragments of the desired transformation construct were designed using the open access Primer3 software [32] with complementary overlaps based on the Gibson Assembly cloning method [33]. Primer sequences and respective amplicons are listed in Appendix A. Transformation constructs were assembled by first generating individual flanking fragments to the target gene to modify, and individual fragments containing fungal and bacterial selectable markers. The individual pieces were ligated in a single Gibson reaction and the resulting fragment was used for transformation of *E. coli*. 

For the development of the *aox* deletion construct, the primer pair oHM39/40 was used to generate a single fragment containing both an ampicillin resistance cassette and an origin of replication element using pUC19 (New England Biolabs, Ipswich, MA, USA) DNA as template. Subsequently, the primer pairs oHM35/36 and oHM37/38 were used to generate upstream and downstream flanking regions, respectively, to *aox* using SRZ2 genomic DNA as template. The primer pair oHM33/34 was used to amplify the hygromycin resistance cassette from pUMA1507 [34] DNA. These fragments were then ligated in a single Gibson reaction and the resulting product was used for transformation of *E. coli*.

The final plasmid was then isolated and linearized with EcoRV (New England Biolabs, Ipswich, MA, USA). Subsequently, the primer pair oHM36/38 was used to amplify the *aox* deletion construct (4582 bp), made up of the flanking upstream region, followed by the hygromycin resistance cassette and the flanking downstream region. The amplified construct was used in the transformation of different strains of SRZ facilitated by homologous recombination. Preliminary verification of deletion mutants was achieved by polymerase chain reaction (PCR) using the primer pair oHM33/68, which resulted in the synthesis of a 4582 bp fragment in successful *aox* deletion mutants. Wildtype (WT) strains were simultaneously used as controls, in which production of a 4005 bp fragment corresponded to the intact *aox* locus.

For the development of the *aox* complementation construct, the primer pair oHM40/146 was used to generate a single fragment containing both the ampicillin resistance cassette and origin of replication element using pUC19 (New England Biolabs, Ipswich, MA, USA) DNA as template. The primer pairs oHM147/148 and oHM39/155 were used to generate upstream and downstream flanking regions, respectively, to *aox* using SRZ2 genomic DNA as template (in this second construct, the upstream region contains Aox in its entirety). The primer pair oHM149/156 was used to amplify the carboxin resistance cassette fragment from pMF4-1c [35] DNA. These fragments were then ligated in a single Gibson reaction and the resulting product was used for transformation of *E. coli*. 

The final plasmid was then isolated and linearized with EcoRV (New England Biolabs, Ipswich, MA, USA). Subsequently, the primer pair oHM30/147 was used to amplify the *aox* complementation construct (5670 bp), made up of the flanking upstream region containing *aox* in its entirety, followed by the carboxin resistance cassette and the downstream flanking region. The amplified construct was used in the transformation of *aox* deletion mutants facilitated by homologous recombination. Preliminary verification of successful transformants was achieved by PCR using the primer pair oHM146/156, which resulted in the production of a fragment 4400 bp in size (WT strains were used as negative controls).

For the investigation of mitochondrial localization of Aox, a fusion construct with a fluorescent protein was designed. The primer pair oHM39/40 was used to generate a single fragment containing both the ampicillin resistance cassette and origin of replication element using pUC19 DNA as template. The primer pairs oHM70/104 and oHM37/38 were used to generate upstream and downstream flanking regions, respectively, to *aox*, using SRZ2 genomic DNA as template (in this third construct, the upstream region contains the *aox* coding sequence without the STOP codon). The primer pair oHM102/103 was used to generate a fragment containing eGFP from pMF4-1c [35] DNA. The primer pair oHM34/105 was used to generate the hygromycin resistance cassette from pUMA1507 [34] DNA. These fragments were then ligated in a single Gibson reaction and the resulting product was used for transfomation of *E. coli*. 

The resulting plasmid was isolated and linearized with EcoRV (New England Biolabs, Ipswich, MA, USA). Subsequently, the primer pair oHM38/70 was used to amplify the *aox*-eGFP fusion construct (6835 bp), made up of the flanking upstream region containg the *aox* sequence without the STOP codon, followed by eGFP fragment, the hygromycin cassette and the flanking downstream region. The amplified construct was used in the transformation of different strains of SRZ facilitated by homologous recombination. Preliminary verification of successful transformants was achieved by PCR using the primer pair oHM34/102, which resulted in the production of a 3101 bp fragment in succesful transformants (WT strains were used as negative controls.

PCR was carried out in a T100 Thermal Cycler (Bio-Rad Laboratories, Hercules, CA, USA) with Ex Taq Hot Start DNA Polymerase (TaKaRa Bio USA, Inc., San Jose, CA, USA), PrimeSTAR Max DNA Polymerase (TaKaRa Bio USA, Inc., San Jose, CA, USA) or DreamTaq Hot Start PCR Master Mix (Thermo Fisher Scientific, Waltham, MA, USA). PCR cycling conditions for Hot Start DNA polymerases involved an initial denaturation step at 94 °C for 4 min, followed by 34 cycles of a three-step process (denaturation at 94 °C for 30 s, annealing at 58–62 °C for 30 s and extension at 72 °C for 1 min per 1 kb of anticipated product) and a final extension step at 72 °C for 10 min. PCR cycling conditions for PrimeSTAR Max DNA Polymerase involved an initial denaturation step at 98 °C for 2 min, followed by 34 cycles of a three-step process (denaturation at 98 °C for 10 s, annealing at 58–62 °C for 15 s and extension at 72 °C for 30–45 s) and a final extension at 72 °C for 5 min. In preparation for Gibson Assembly, all PCR products were generated using PrimeSTAR Max DNA Polymerase. Gibson Assembly was carried out using the Gibson Assembly Master Mix kit (New England Biolabs, Ipswich, MA, USA) and according to the manufacturer’s instructions.

### 2.4. Growth Inhibition Assays

Haploid SRZ strains were grown overnight in 4 mL of PD broth on a rotary shaker at 200 rpm at 28 °C. Cells were harvested by centrifugation at 14,000 rpm, quantified using a hemocytometer and adjusted with fresh PD broth, supplemented with antimycin A (AMA; 50 µM) and/or sailcylhydroxamic acid (SHAM; 2 mM), to 10^5^ cells/mL. Control treatments included PD broth supplemented with the respective respiratory inhibitor solvent (ethanol for AMA, and DMSO for SHAM). Cells were incubated overnight on a rotary shaker at 200 rpm at 28 °C and subsequently sub-cultured onto PD agar for determination of total viable colonies (TVC) per mL of medium. One-way ANOVA followed by Tukey’s Multiple Comparison test was performed in GraphPad Prism 9.2.0 (GraphPad Software, LLC, San Diego, CA, USA) to determine statistical significance, with *p* < 0.05 = significant and *p* > 0.05 = not significant.

### 2.5. Pathogenicity Assays

Strains were grown in 50 mL of PD broth on a rotary shaker at 28 °C until an OD_600_ of 0.5–0.8 was reached. Cells were harvested by centrifugation at 3500 rpm and resuspended in ddH_2_O to a final OD_600_ of 2. The desired strain combinations were then mixed in a 1:1 ratio and used to inoculate 7-day old Tom Thumb maize (High Morning Organic Seeds, Wolcott, VT, USA) seedlings. Disease evaluation was performed 7–8 weeks post infection (wpi), in which plant height was recorded and symptoms were scored for the calculation of disease incidence indexes and according to preestablished criteria [4]. One-way ANOVA followed by Tukey’s Multiple Comparison test was performed in GraphPad Prism 9.2.0 (GraphPad Software, LLC, San Diego, CA, USA) to determine statistical significance, with *p* < 0.05 = significant and *p* > 0.05 = not significant

### 2.6. Expression Studies

Two-step reverse transcription quantitative PCR (RT-qPCR) was used for the verification of gene deletions and differential gene expression analysis. Nucleic acids were isolated from SRZ haploid sporidia, diploid teliospores and mated cells by homogenization in a frozen mortar and pestle and treatment with TRIzol reagent (Invitrogen, Waltham, MA, USA). Following cell disruption, RNA was isolated using the Direct-zol RNA Miniprep kit (Zymo, Irvine, CA, USA). Subsequent cDNA synthesis was carried out using Super Script IV (Invitrogen, Waltham, MA, USA) and followed the manufacturer’s instructions. qPCR was carried out in a StepOne Real-Time PCR System (Applied Biosystems, Waltham, MA, USA) using EvaGreen dye (Mango Biotechnology, Cambridge, MA, USA) following the manufacturer’s instructions. Cycling conditions involved an initial denaturation step at 95 °C for 10 min, followed by 40 cycles of 95 °C for 15 s and 60 °C for 1 min. 

Melting curve analysis was performed at the end of each cycle to ensure specificity of the reaction. Gene-specific primers were designed using the open access Primer3 software [32] and are included in Appendix A. The glyceraldehyde 3-phosphate dehydrogenase (*gapdh*, sr109402) was used as endogenous control for the calculation of relative expression levels using the 2^−ΔΔCt^ (Livak) method [36]. Ct values above 30 or not detectable were indicators of no gene expression. The Student’s t-test was performed in GraphPad Prism 9.2.0 (GraphPad Software, LLC, San Diego, CA, USA) to determine statistical significance, with *p* < 0.05 = significant and *p* > 0.05 = not significant.

### 2.7. Microscopy

Fluorescence images were acquired using a Nikon A1R confocal microscope. Transformed SRZ strains were grown overnight at 28 °C in PD broth to an OD_600_ of 1. Ten-fold dilutions of each culture were made and 500 µL were placed in the wells of 35-mm CELLview culture dishes (Greiner Bio-One, Monroe, NC, USA). eGFP signal was acquired using a 488 nm line of an Argon laser and was detected at 510 nm. A transmitted light image was acquired during scanning for visualization of cell outline by grouping of the transmitted detector with the argon laser. For confirmation of localization, cells were treated with 100 µM MitoView Blue (Biotium, Fremont, CA 94538, USA) for 10 min. Blue fluorescence was acquired using the DAPI detection channel and overlapped with eGFP fluorescence.

## 3. Results

### 3.1. Identification of a Putative Alternative Oxidase in SRZ

The amino acid sequence of the *U. maydis* Aox1 was used in a BLASTp search against the nuclear genome of SRZ. The resulting sequence of 409 amino acids was then used for MUSCLE analysis against known alternative oxidases in closely and distantly related species (Figure 1). Protein identities of the putative Aox of SRZ were 85.02% for *S. scitamineum*, 85.48% for *S. graminicola*, 67.06% for *U. bromivora*, 70.91% for *U. maydis* and 72.22% for P. hubeiensis. The highest amino acid identities were between the closest relatives of SRZ (*S. scitamineum* and *S. graminicola*) and were consistent with the sizes of their predicted polypeptides (402 and 412 amino acids, respectively). Greater size discrepancies were seen in *U. maydis* (448 amino acids), *U. bromivora* (417 amino acids) and, more notably, in the distantly related *P. hubeiensis* (323 amino acids). Amino acid conservation was strongest in the carboxyl end of the aligned polypeptide sequences, coinciding with a ferritin-like diiron-binding domain (cd01053, E-value = 2.43^−^^85^) common in alternative oxidases of plants, fungi and protists. This preliminary bioinformatic evidence strongly suggests the presence of an aox gene in SRZ, encoding a polypeptide of approximately 45 kDa and rich in alanine (13.45%).

### 3.2. SRZ Aox Localizes to the Mitochondrial Membrane

To determine whether the SRZ Aox ortholog identified bioinformatically corresponds to a mitochondrial protein, a fluorescent Aox-eGFP fusion protein was designed. The SRZ transformants generated with this construct were then analyzed using confocal microscopy. Fluorescence was detected as thin threads that spread out throughout the cell, and even extended to budding daughter cells (Figure 2). 

To further confirm that the fluorescence detected originated from mitochondrial structures, cells were treated with the mitochondrial fluorogenic dye MitoView Blue. This chemical has light absorbance and emission at 398 nm and 440 nm, respectively, and can permeate membranes. Upon accumulation in the mitochondrial membrane, the chemical becomes brightly fluorescent. Treatment of cells with this mitochondrial fluorogenic dye confirmed that the green fluorescence coincided with mitochondrial structures (Figure 3).

### 3.3. Aox Provides an Alternative Route for the Transport of Electrons during Oxidative Phosphorylation

To functionally confirm the presence of an alternative oxidase in the ETC of SRZ, cells were grown in the presence of specific respiratory inhibitors. Cells were treated with antimycin A (AMA), salicylhydroxamic acid (SHAM), or both to infer the predominant respiratory pathway as a function of cell survival, with AMA targeting complex III and SHAM targeting Aox. In addition to an untreated control group, cells were treated with the respective solvents used to prepare the respiratory inhibitors to account for their basal toxicity on SRZ cells. Treatment with AMA and/or SHAM resulted in reduced survival rate in WT strains, although it did not eliminate growth (Figure 4). A significantly higher number of cells was viable after treatment with SHAM alone than with AMA alone, and growth increase was smallest when cells were treated with both AMA and SHAM (Figure 4). This suggests that SRZ is relatively sensitive to AMA and, to a lesser degree, to SHAM. This finding suggests that cells can survive despite the inhibitory action of AMA on the standard cytochrome c pathway by virtue of an alternative respiratory pathway promoted by Aox.

We next tested the effect of the respiration inhibitory substances AMA and SHAM on the growth of the generated *aox* deletion strains. Addition of AMA and of AMA and SHAM, but not of SHAM alone had a severe growth inhibitory effect on the deletion strains (Figure 4). AMA had a more serious effect on the mutants than it did on the WT strains, suggesting that in *aox* deletion strains, movement of electrons is predominantly via the cytochrome c pathway. 

Lastly, complementation of the deleted *aox* gene was achieved by transformation of protoplasts of the deletion mutant strains with the complementation construct. This construct incorporated the *aox* gene in its entirety as part of the upstream flank, in addition to the extra DNA sequence flanking the insertion target site. Successful homologous recombination resulted in the replacement of the hygromycin resistance cassette used to disrupt *aox* by a carboxin resistance cassette and the *aox* gene in its native state. The resulting complemented strains were subjected to the same inhibitory treatment as the WT strains. Complementation successfully restored respiratory capacity of *aox* deletion strains in presence of AMA (Appendix A).

### 3.4. Alternative Respiration Is Involved in Pathogenicity of SRZ

To determine whether Aox is a crucial component of SRZ during its pathogenic stage, maize seedlings were inoculated with compatible combinations of both WT and *aox* mutant strains. Symptomatic profiles were analyzed once plants were fully matured (7–8 weeks post inoculation) and are illustrated in Figure 5a. Interestingly, all symptoms were present in plants infected by both WT and mutant strains. However, plants inoculated with the *aox* deletion strains displayed a reduction in serious symptoms (tassel with spores, leafy tassel and ear with spores) compared to plants inoculated with the WT strains. This last finding was discernable in the overall disease incidence analysis, in which individual symptoms were taken into consideration to ascribe single ratings to the different symptomatic profiles generated. Along these lines, mating combinations of the WT strains were responsible for significantly higher disease incidence indexes than those caused by combinations between the *aox* mutants. The mutants’ reduced disease incidence without fully disrupting the fungal life cycle may indicate that Aox is acting synergistically with other cellular components during the pathogenic stage of SRZ.

### 3.5. Aox Expression Is Upregulated during Teliospore Stage of Fungal Life Cycle

To determine whether Aox is involved in developmental regulation, gene expression analysis was implemented. Accordingly, *aox* transcriptional expression was compared via qRT-PCR during the teliospore (diploid), mated (beginning of dikaryon phase) and sporidium (haploid) stages of SRZ. Relative expression analysis of *aox* in teliospores was performed separately in reference to SRZ1 (Figure 6) and SRZ2 (Appendix A) sporidia. As controls, teliospores resulting from a compatible cross between *aox* deletion mutants, as well as mated cells resulting from combining compatible haploid sporidia, were included. In *aox* deletion strains, no expression was observed. In WT strains, *aox* was found not to be upregulated in mated cells but be upregulated in teliospores (Figure 6, Appendix A). These results suggest that Aox-mediated respiration is favored during the teliospore developmental stage, consistent with the lower metabolic demands of a quiescent life phase. These findings, however, should be subject of further experimentation to confirm that the higher amounts of Aox transcript detected in teliospores coincides with higher amount of active Aox and higher respiratory capacity.

## 4. Discussion

Identification and characterization of alternative components of the ETC recently received special attention across different fungal groups, linking them to thermotolerance, pathogenicity and the overall cellular welfare in the face of chemical and environmental stresses. In some cases, however, the benefit of alternative respiration remains unclear. For instance, studies in *U. maydis* were inconclusive regarding when the fungus switches respiratory routes at different stages of its life cycle. Accordingly, the current study focused on identifying an Aox homolog in the related species SRZ, as it provides some peculiarities during its life cycle that may shed light on the advantages of Aox-mediated respiration. The bioinformatic analysis used for the identification of Aox in SRZ revealed high amino acid identity to that encoded by the *U. maydis* ortholog, Aox1. The analysis also included Aox amino acid sequences of species closely and distantly related to SRZ, providing further support for the identification of a putative Aox in SRZ. Despite some polypeptide size differences between the predicted Aox polypeptide of SRZ and that of distantly related species, amino acid identity was relatively high across all species. More importantly, the analysis revealed the presence of an iron binding domain in all sequences, that was previously characterized in other organisms [16,17].

Functional verification of the putative Aox was based on the effect of mitochondrial inhibitors on overall cell growth. This assay proved to be uncomplicated and cost-effective, as it permitted the direct assessment of the effects of specific mitochondrial inhibitors on cell survival. The classical cytochrome c pathway (Figure 7) was blocked by inhibiting cytochrome bc_1_ (complex III) using AMA. AMA is a bacterial secondary metabolite that acts by binding to the Qi site of complex III, thereby competing with coenzyme Q to arrest the movement of electrons. Ultimately, this blockage leads to electron efflux, with the potential to become harmful as electrons leak into mitochondrial spaces to form highly toxic products, such as reactive oxygen species (ROS).

The presence of an alternative oxidase provides an alternate route for electron transport, receiving electrons directly from reduced ubiquinol to reduce oxygen into water. This one-step process only depends on NADH oxidoreductase (complex I) as the sole contributor to the proton motive force in the intermembrane space of the mitochondrion. An organism equipped with this diverging route for electron transport can dissipate the build-up of electrons caused by AMA and survive the inhibition of the classical cytochrome c pathway, albeit with a significant decrease in ATP synthesis.

A loss-of-function approach suggested notable differences during the pathogenic stage of SRZ. Although WT and mutant crosses of SRZ successfully colonized, invaded and induced symptoms in the plant, they did so at significantly different frequencies and deletion of *aox* resulted in overall weaker symptoms. Deletion of *aox* in *U. maydis* did not result in weaker virulence. However, *U. maydis* only produces local infections, and symptoms are scored as early as 2 wpi. SRZ infection provides the ability for long-term evaluation of the phytopathogen, as disease evaluation requires fully mature maize plants. Accordingly, Aox may be involved in the endurance of the fungus once inside of the host plant. The controlled nature of the infection experiments possibly contributed to the survival of the deletion mutants in the plant tissue, as a natural population of maize is constantly challenged by environmental insults. Further experimentation is required and should take into consideration environmental factors that may affect both the fungus and the host plant, such as extreme temperatures, to uncover additional aspects of the involvement of Aox in pathogenicity.

Expression analysis of different stages of the life cycle of SRZ revealed that *aox* is upregulated in teliospores, in contrast with haploid sporidia and even mated cells. The analysis was performed in reference to different haploid strains, as these behaved differently in the conducted qRT-PCR experiments. However, *aox* transcript levels of teliospores remained significantly higher regardless of the reference strain used. This finding is consistent with what was found in other organisms, in which specific life cycle stages favor secondary respiration in response to different metabolic demands. This finding suggests that Aox may play an important role in the teliospore stage of SRZ. Additional experimentation is needed to confirm if these specialized developmental structures, which remain relatively quiescent until they disseminate and germinate, do in fact have increased respiration rates mediated by Aox. In contrast, given that haploid sporidia are in constant division and must grow by budding or, after successful mating, filamentously for successful pathogenic development, ATP requirements are significantly higher. The higher metabolic demand imposed by sporidia can thus be fulfilled by the canonical cytochrome c pathway of the ETC, with three of the four classical protein complexes contributing to the mitochondrial proton motive force.

The presence of Aox in plants contributes additional complexities to studies of this nature, as the evolutionary history of host-pathogen interactions should not be ignored. Quite surprisingly, the Aox of *Zea mays* (Aox1a, NCBI Accession No. LOC100273671) that was identified and extensively characterized early, was linked to thermotolerance and osmotic and oxidative regulation [37,38,39]. Along these lines, many arguments could be made regarding the biological interactions between SRZ and maize throughout their evolutionary history that may explain the emergence of Aox as an adaptive innovation or as simply the result of divergent evolution.

## Figures and Tables

**Figure 1 jof-08-00148-f001:**
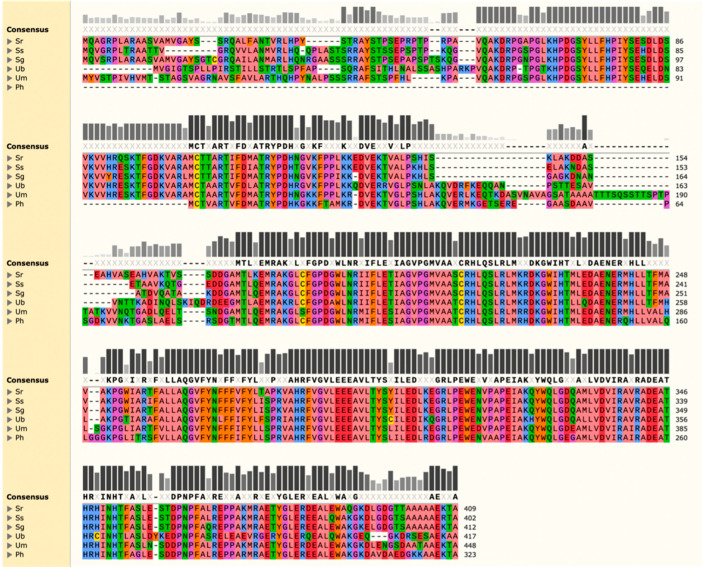
Multiple Sequence Comparison by Log-Expectation (MUSCLE) analysis of amino acid sequence of putative *S. reilianum* f. sp. *zeae* (SRZ) Aox against known polypeptides. Aox amino acid sequences of the following organisms were used: *S.*
*scitamineum* (Ss, CDU22616.1), *S. graminicola* (Sg, XP_029737873.1), *U. bromivora* (Ub, SAM82122.1), *U. maydis* (Um, XP_011389130.1) and *P. hubeiensis* (Ph, XP_012186999.1). Amino acids with similar physico-chemical properties are represented by the same color. Bars represent amino acid conservation across all sequences analyzed, with darker and taller bars corresponding to higher conservation indexes. Amino acids in bold make up the consensus sequence and represent 100% conservation in that position.

**Figure 2 jof-08-00148-f002:**
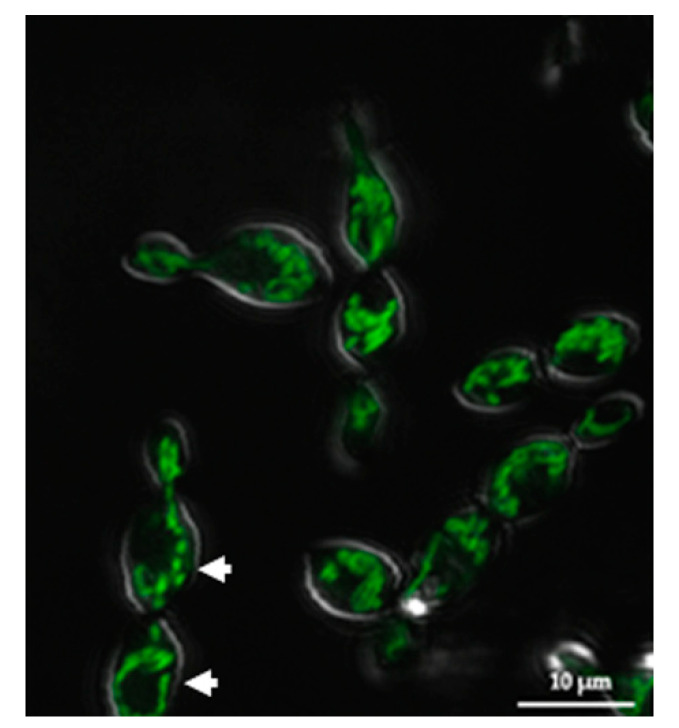
Confocal microscopy of cells transformed with the Aox-eGFP fusion construct. The Aox-eGFP protein fusion renders mitochondrial structures (arrowheads) as thin threads within budding cells.

**Figure 3 jof-08-00148-f003:**
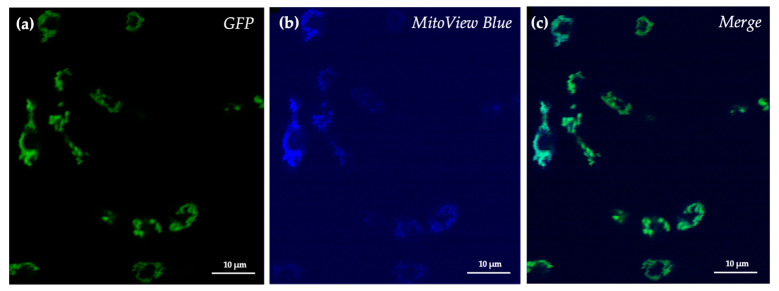
Confirmation of localization of Aox in SRZ. (**a**) eGFP fluorescence. (**b**) MitoView Blue fluorescence. (**c**) Merged green and blue fluorescence.

**Figure 4 jof-08-00148-f004:**
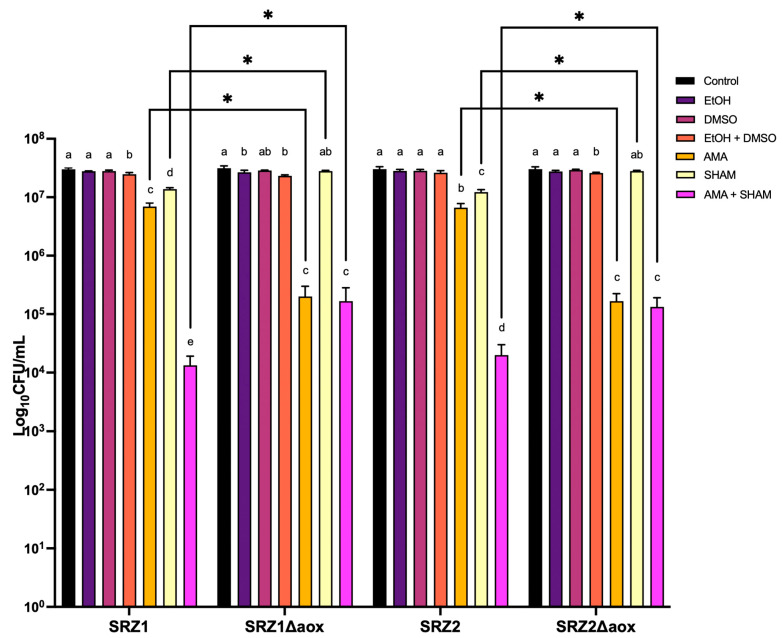
Growth inhibition assay of SRZ. A total of 10^5^ cells/mL were treated as indicated in the figure key and incubated at 28 °C for 24 h. Control groups consisted of untreated cells grown in PD broth. AMA and SHAM were used at concentrations of 50 µM and 2 mM, respectively. Cultures were then plated onto PD agar to determine number of surviving colonies. Bars represent averages of biological triplicates with standard errors indicated. One-way ANOVA followed by Tukey’s Multiple Comparison Test was performed in Graphpad 9.0. Letters above bars represent significant differences (*p* < 0.05) between different treatments in reference to corresponding control group of each strain. Comparisons of treatments between different strains are indicated by connecting black line brackets and significant differences (*p* < 0.05) are represented by asterisks.

**Figure 5 jof-08-00148-f005:**
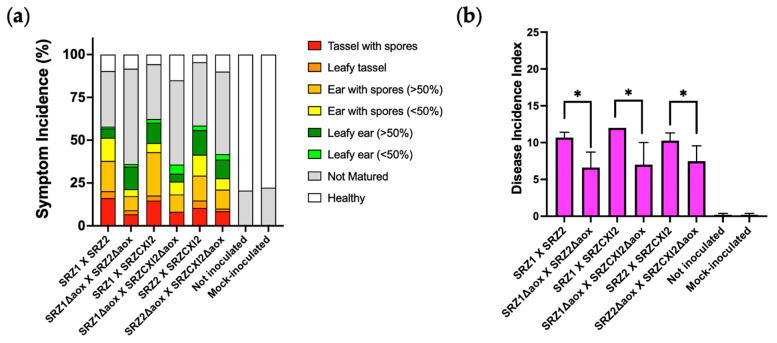
Assessment of pathogenicity of SRZ on maize. (**a**) Mutant and WT strains of SRZ were used to inoculate maize seedlings. Inoculations were performed in triplicate, with 20 plants per replicate (N = 60). Symptoms were scored at 7–8 wpi according to preestablished criteria [4]. (**b**) Mean disease incidence indexes were calculated based on the individual symptoms ranking. Bars represent averages of biological triplicates, with standard errors indicated. Comparisons were made between WT crosses and their corresponding *aox* deletion mutant crosses, with significant differences indicated by asterisks above bars (*p* < 0.05).

**Figure 6 jof-08-00148-f006:**
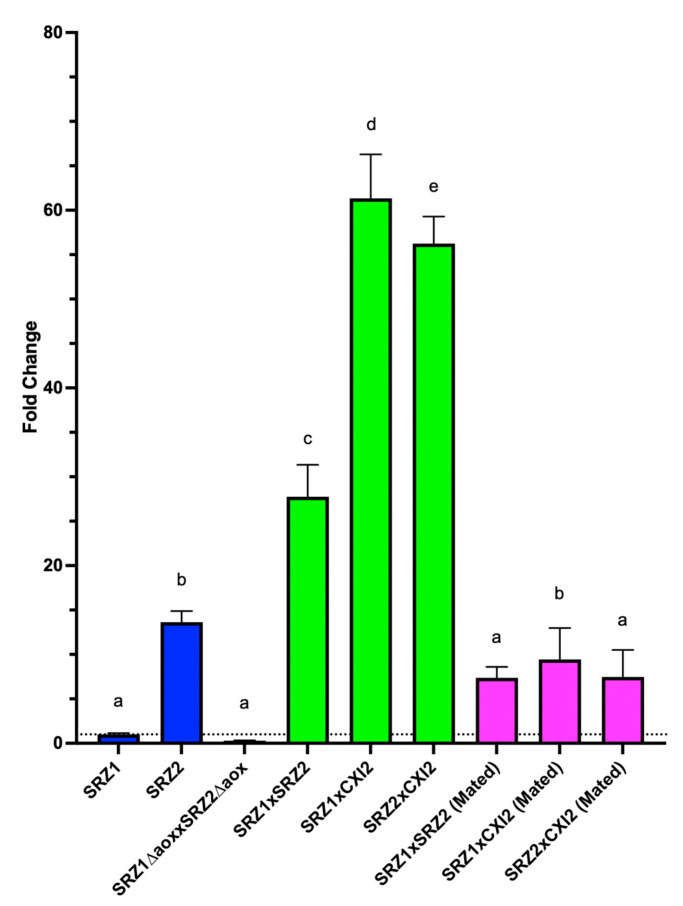
Fold change differences of *aox* expression in teliospores and haploid cells of SRZ in reference to SRZ1. Analysis was performed relative to *gapdh* expression as endogenous control. Relative expression levels were calculated using the 2^−ΔΔCt^ method and were performed in reference to SRZ1. Mated cells and teliospores produced in a cross between *aox* deletion mutants were included as controls. Coloring of bars corresponds to cell type: blue = haploid cells, green = teliospores, pink = mated cells. Significant differences are indicated by letters on top of bars (*p* < 0.05).

**Figure 7 jof-08-00148-f007:**
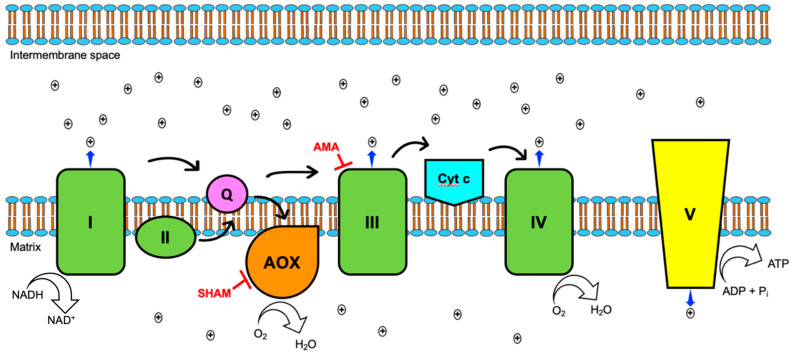
Aox-mediated respiration. Black arrows indicate movement of electrons. The canonical cytochrome c pathway includes the classical protein complexes, depicted in green, with protons pumped into the intermembrane space by complexes I, III and IV. This pathway is coupled to ATP synthesis by contributing to the proton gradient in the intermembrane space that will drive ATP synthesis via ATP synthase (complex V, in yellow). Inhibition of complex III by drugs like anti-mycin A (AMA) halts movement of electrons via the classical route. Aox (in orange) functions as complex IV, reducing oxygen into water, thus providing an additional route to finalize electron flow. However, Aox does not function as a proton pump, effectively diminishing the proton gradient and, consequently, reducing ATP synthesis. Aox may be targeted for inhibition by chemicals like salicylhydroxamic acid (SHAM).

## Data Availability

Gene and protein sequences are available at NCBI.

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
