# Peer review of "Identification and Functional Characterization of a Putative Alternative Oxidase (Aox) in Sporisorium reilianum f. sp. zeae"

_jof, 2022, doi:10.3390/jof8020148_

Round 1
Reviewer 1 Report
Comments for authors
In this manuscript, the authors characterized the alternative oxidase (Aox) in Sporisorium reilianum f. sp. zeae (SRZ). First, the authors identified Aox gene in SRZ by homology search and sequence alignment. Co-localization of Aox-eGFP and MitoView Blue indicates Aox as a protein in mitochondria. Then, the authors performed growth inhibition assay on WT, aox mutant, and genetic complementation strains. They found that AMA, an inhibitor on the protein complex III of the classical cytochrome c pathway, greatly reduced growth on aox mutants compared to WT and genetic complementation strains. However, SHAM, an inhibitor on alternative oxidases, did not have a growth inhibition effect on aox mutants. This confirms the function of the identified gene in SRZ as alternative oxidase. The infection assay found that aox mutants have reduced virulence compared to WT strains. Finally, the expression analysis showed that Aox expression is increased in a teliospore stage compared to other stages of a life cycle, suggesting a role of Aox during a quiescent stage on a plant host.
I think the manuscript is well written and easy to follow. All experiments and interpretation are cohesive, and discussion is scientifically reasonable. I have a few comments to improve the manuscript quality. Please see below for my comments.
Major comments
- I think Figure S1 is also a key to prove the functionality of Aox, and this should be incorporate as a part of Figure 4. This is to rule out a possibility that altered phenotype is due to Aox1, but not other side effects from genetic manipulation. The authors should also make a statistical comparison of three strains (WT, aox mutant, and complementation strains) among the AMA and/or SHAM treatments. This is to clearly illustrate that aox knockout results in reduced growth compared to WT under the AMA treatment, and genetic rescue reverts the growth to the WT level. Same thing applies for the SHAM treatment. The authors can cluster bars from each strain together to save space.
- I personally discourage people presenting data with several p-values to show the level of significance. This not only provides an inadequate way to represent statistical difference among groups/treatments, but also distorts the meaning of p-value in hypothesis testing. Instead, I recommend the authors using letters to assign statistically different groups from the Tukey’s post-hoc test. See an example in the Word file, which I provide ‘a’ as a statistical group with a higher value than ‘b’, and so on. Asterisks can also be used to compare certain treatments among different strains. This can be applied for Figures 4, 5, 6, S1 and S2.
- I wonder if the authors tested the infection assay using complementation strains, or even making a hybrid cross between Aox WT and Aox mutant. This will be an interesting to see if Aox function in a half dosage provides a similar virulence level to the WT cross.
Minor comments
- Line 42: I think it is a good idea to spell out the formae speciales, like “with two formae speciales, S. reilianum f. sp. zeae (SRZ) that infects maize and S. reilianum f. sp. sorghi (SRS) that infects sorghum.” This would link to the title that the authors spelled out the full speciales name for SRZ.
- Line 58: Should the authors count FADH2 as another electron donor too?
- Line 60ff: I think it is a good idea to briefly mention what are classical protein complexes I-IV in the previous paragraph, so that it will link to this statement when the authors mention “the classical cytochrome c pathway”.
- Line 64: Please consider changing to ‘potassium cyanide (KCN)’ or ‘cyanide (CN-)’.
- Line 75 – 79: Please consider paraphrasing this sentence as it is little hard for readers to understand.
- Line 98: Increased expression of what? Aox? Please specify.
- Line 254: Please italicize S. scitamineum.
- Figure 1: In the legend, please specify species abbreviations used in the alignment. Like Sr is for S. reilianum, Ss is for S. scitamineum, and so on. Alternatively, the authors can use sequence accessions or spell out full species names. Also, it would be nice to highlight the ‘ferritin-like diiron-binding domain’ region in the alignment. Having additional sequences from plants and yeasts in the same alignment would also provide strong evidence for the high conservation of this domain.
- Figures 2 – 3: I think these two figures can be merged as one. For Figure 2, I think it is a good idea to provide separate panels for bright field and GFP, similar way with Figure 3. Using arrowheads to indicate ‘mitochondrial structures as thin threads’ will be a good visual aid for readers. Please consider improving graphic quality for Figure 3. It is hard for me to see overlapping color in the merge panel.
- Line 281ff: I think the authors should clarify here that AMA and SHAM inhibit which protein complexes, and which inhibitor is a key to determine the function of alternative oxidase. Making a good clarification at the beginning of the paragraph will help readers understand results and data interpretation.
- Line 298: I am not sure if ‘unidirectional’ is a proper word here. When I first read this, I would interpret that then the WT strains have ‘bidirectional electron transport’. However, electron transport in the WT strains flow in two different routes (through classical cytochrome c and Aox), but not two opposite directions as the word ‘bidirectional’ means. I recommend changing words to something like ‘movement of electrons is only through the classical cytochrome c pathway’.
- Figure 4 caption: Please specify if these cells were treated under PDB?
- Line 363: I would say ‘fungal groups’ instead of ‘families’, as Aox have been studied in various classes and phyla of fungi.
- Line 376 – 377: Again. Including sequences from other organisms in the alignment will provide strong evidence of conserved iron binding domain.
- Figure 7: In the Figure, please add electron transport from NADH to protein complex I. Please indicate a symbol for a proton (a circle with a plus sign). Please consider improving the graphic quality.
- I suggest the authors to write a concluding paragraph at the very end to summarize all findings and a brief implication from the study, but this is up to the authors.
Author Response
Reviewer 1
Comments for authors
In this manuscript, the authors characterized the alternative oxidase (Aox) in Sporisorium reilianum f. sp. zeae (SRZ). First, the authors identified Aox gene in SRZ by homology search and sequence alignment. Co-localization of Aox-eGFP and MitoView Blue indicates Aox as a protein in mitochondria. Then, the authors performed growth inhibition assay on WT, aox mutant, and genetic complementation strains. They found that AMA, an inhibitor on the protein complex III of the classical cytochrome c pathway, greatly reduced growth on aox mutants compared to WT and genetic complementation strains. However, SHAM, an inhibitor on alternative oxidases, did not have a growth inhibition effect on aox mutants. This confirms the function of the identified gene in SRZ as alternative oxidase. The infection assay found that aox mutants have reduced virulence compared to WT strains. Finally, the expression analysis showed that Aox expression is increased in a teliospore stage compared to other stages of a life cycle, suggesting a role of Aox during a quiescent stage on a plant host.
I think the manuscript is well written and easy to follow. All experiments and interpretation are cohesive, and discussion is scientifically reasonable. I have a few comments to improve the manuscript quality.
We thank the Reviewer for their thorough reading of the paper and for their constructive comments and suggestions for improvement.
Please see below for my comments.
Major comments
- I think Figure S1 is also a key to prove the functionality of Aox, and this should be incorporate as a part of Figure 4. This is to rule out a possibility that altered phenotype is due to Aox1, but not other side effects from genetic manipulation. The authors should also make a statistical comparison of three strains (WT, aox mutant, and complementation strains) among the AMA and/or SHAM treatments. This is to clearly illustrate that aox knockout results in reduced growth compared to WT under the AMA treatment, and genetic rescue reverts the growth to the WT level. Same thing applies for the SHAM treatment. The authors can cluster bars from each strain together to save space.
- I personally discourage people presenting data with several p-values to show the level of significance. This not only provides an inadequate way to represent statistical difference among groups/treatments, but also distorts the meaning of p-value in hypothesis testing. Instead, I recommend the authors using letters to assign statistically different groups from the Tukey’s post-hoc test. See an example in the Word file, which I provide ‘a’ as a statistical group with a higher value than ‘b’, and so on. Asterisks can also be used to compare certain treatments among different strains. This can be applied for Figures 4, 5, 6, S1 and S2.
We thank the Reviewer for this suggestion. We have modified figures as suggested, although we respectfully disagree with the suggestion to merge the data in Fig. S1 with those in Fig. 4. So doing would visually be difficult for the reader due to size constraints. Instead, we have added the comparisons of WT vs the complemented strains in Fig. S1 to address the concerns of the Reviewer. Importantly, the complemented strains are rescued for their sensitivity to SHAM, equivalent to that of WT.
- I wonder if the authors tested the infection assay using complementation strains, or even making a hybrid cross between Aox WT and Aox mutant. This will be an interesting to see if Aox function in a half dosage provides a similar virulence level to the WT cross.
We tested this and saw no differences with WT strains in terms of infection. What would be worth looking at later would be the spores resulting from AOX WT, an AOX mutant, and the complemented strains, which we did not get to in this study.
Minor comments
- Line 42: I think it is a good idea to spell out the formae speciales, like “with two formae speciales, S. reilianum f. sp. zeae (SRZ) that infects maize and S. reilianum f. sp. sorghi (SRS) that infects sorghum.” This would link to the title that the authors spelled out the full speciales name for SRZ.
We made this change.
- Line 58: Should the authors count FADH2 as another electron donor too?
The goal of this section was to summarize the major route(s) of the electron transport chain (ETC), without additional inputs from other sources, with the purpose of highlighting a “major” route vs. the “alternate” route. Including in the description FADH2 as an another electron donor, we feel, would dilute that take-home message.
- Line 60ff: I think it is a good idea to briefly mention what are classical protein complexes I-IV in the previous paragraph, so that it will link to this statement when the authors mention “the classical cytochrome c pathway”.
Thank you for this suggestion, we have made this change.
- Line 64: Please consider changing to ‘potassium cyanide (KCN)’ or ‘cyanide (CN-)’.
We have changed the wording to “potassium cyanide.”
- Line 75 – 79: Please consider paraphrasing this sentence as it is little hard for readers to understand.
We have re-worded this section as follows to make it more clear: “…alternative oxidase (Aox), which functions like complex IV and catalyzes the reduction of oxygen into water but does not function as a proton pump [7]. In this manner, electron flow is limited to complexes I and II, with complex I being the sole contributor to the electrochemical gradient in the mitochondrial intermembrane space.”
- Line 98: Increased expression of what? Aox? Please specify.
We now specify that we are referring to “increased aox transcript levels”
- Line 254: Please italicize S. scitamineum.
Change made.
- Figure 1: In the legend, please specify species abbreviations used in the alignment. Like Sr is for S. reilianum, Ss is for S. scitamineum, and so on. Alternatively, the authors can use sequence accessions or spell out full species names. Also, it would be nice to highlight the ‘ferritin-like diiron-binding domain’ region in the alignment. Having additional sequences from plants and yeasts in the same alignment would also provide strong evidence for the high conservation of this domain.
- Figures 2 – 3: I think these two figures can be merged as one. For Figure 2, I think it is a good idea to provide separate panels for bright field and GFP, similar way with Figure 3. Using arrowheads to indicate ‘mitochondrial structures as thin threads’ will be a good visual aid for readers. Please consider improving graphic quality for Figure 3. It is hard for me to see overlapping color in the merge panel.
We did not want to merge Fig. 2 and Fig. 3, in part because we did not have a brightfield image for Fig. 2, and yet, the image we do have is very clear and illustrates the movement of the mitochondria in dividing cells. We added arrowheads to indicate the mt structures.
- Line 281ff: I think the authors should clarify here that AMA and SHAM inhibit which protein complexes, and which inhibitor is a key to determine the function of alternative oxidase. Making a good clarification at the beginning of the paragraph will help readers understand results and data interpretation.
We now clearly indicate which protein complexes are inhibited by AMA and by SHAM.
- Line 298: I am not sure if ‘unidirectional’ is a proper word here. When I first read this, I would interpret that then the WT strains have ‘bidirectional electron transport’. However, electron transport in the WT strains flow in two different routes (through classical cytochrome c and Aox), but not two opposite directions as the word ‘bidirectional’ means. I recommend changing words to something like ‘movement of electrons is only through the classical cytochrome c pathway’.
We have made the suggested wording change.
- Figure 4 caption: Please specify if these cells were treated under PDB?
It is now explicitly stated in the legend that cells were grown in PDB.
- Line 363: I would say ‘fungal groups’ instead of ‘families’, as Aox have been studied in various classes and phyla of fungi.
We have made the suggested change in wording.
- Line 376 – 377: Again. Including sequences from other organisms in the alignment will provide strong evidence of conserved iron binding domain.
We respectfully decline to follow this suggestion, as the current alignments sufficiently indicate that Aox from S. reilianum is likely an alternative oxidase.
- Figure 7: In the Figure, please add electron transport from NADH to protein complex I. Please indicate a symbol for a proton (a circle with a plus sign). Please consider improving the graphic quality.
We have added the electron transport from NADH to Complex I and increased the dpi of the figure.
- I suggest the authors to write a concluding paragraph at the very end to summarize all findings and a brief implication from the study, but this is up to the authors.
We thank the Reviewer for this suggestion, but feel that the Abstract and Discussion highlight the primary findings and their relative importance and implications.
Reviewer 2 Report
This work describes the generation of an AOX deletion mutant of the phytopathogenic basidiomycete Sporisorium reilianum and its effect on growth and its involvement in the pathogenic capacity of the fungus. The results show that in the absence of the AOX the serious symptoms of the plant infection are reduced. Expression of the AOX mRNA is higher at the teliospore stage. The manuscript is well written in good English.
Next, I have some minor comments:
line 72: The sentence “the AOX is an inefficient proton pump” is inaccurate and ambiguous, because this enzyme is not a proton pump.
Lines 97-100: Additionally, expression analysis of distinct developmental stages of the fungus indicated increased expression in teliospores, corresponding with the use of a less energy-providing respiratory pathway during quiescent stages of the life cycle.
Is there any data indicating that the cytochrome pathway is missing in the teliospore? Has the respiratory capacity of the AOX and the classic pathway been studied in the teliospore?. Even if the AOX is in higher amounts in the teliospore, the enzyme may be in an inactive state. Using an oxygen isotope discrimination technique in plants, it has been shown that AOX has a minor contribution to respiration, even if its capacity is large (https://doi.org/10.3389/fpls.2021.752795). About 10-30%.
Lines 285-286: Treatment with AMA and/or SHAM resulted in a reduced survival rate in wildtype (WT) strains, although it did not eliminate growth.
I understand that the experimental design in figure 4 measures two things: a) viability after the incubation with the inhibitors and b) growth (observed as an increase in the number of cells after the incubation). So, I agree with the authors that in the presence of AM or SHAM all the strains were capable of growing. However, in the presence of both AM and SHAM, approximately 90% of the WT cells died during the incubation (initial number of cells = 10^5, final number=10^4). Interestingly, this large decrease in cell number only occurred with the WT, but not with the AOX mutants. Fort the mutants, it seems that cells did not grow (maintained the initial number at 10^5), but their viability remained. Is there an explanation?
A second point: According to figure 2, the WT cells have a sufficient amount of AOX. However, the growth of wild-type cells containing the AOX was the same as that of the two AOX deletion mutants in the absence of inhibitors (Figure 4). Should this result indicate that the AOX in the wild-type has no activity or a very small activity during the growth period?
Lines 352-354: These results suggest that Aox-mediated respiration is favored during the teliospore developmental stage, consistent with the lower metabolic demands of a
quiescent life phase
1) I think this proposal is speculative. Is there any data indicating that the cytochrome pathway is missing in the teliospore? Has the respiratory capacity of the AOX and the classic pathway been studied in the teliospore?. Even if the AOX is in higher amounts in the teliospore, the enzyme may be in an inactive state. Using an oxygen isotope discrimination technique in plants, it has been shown that AOX has a minor contribution to respiration, even if its capacity is large. About 10-15%.
2) Despite the presence of an apparently large amount of AOX in the WT (Figure 2), the concentration of mRNA was quite small. In this sense, Nargang reported that an increase in Neurospora AOX mRNA is not reflected in an increase in the protein (https://doi.org/10.1534/g3.119.400522). Following this line of reasoning, it is possible that an increasing the AOX mRNA in the teliospores will not result in a higher content of the protein. The best way to measure this expression is with antibodies against the AOX or the fluorescence of the AOX-gfp.
Figure 6: If the comparison is with SRZ1 and given the standard deviations or standard errors, I see a significant difference for all the mated samples and the SRZ1 condition. On the other hand, the values of the mated samples are between the values of SRZ1 and SRZ2.
Lines 383-385: Antimycin A does not displace ubiquinone or coenzyme Q. Antimycin A competes with ubiquinone for the Qi site; therefore, it prevents the binding of ubiquinone to this site. As a consequence of the strong binding of antimycin A to the Qi site, the flow of electrons associated with the Q cycle stops. This blockage cannot give rise to a flow of electrons. Certainly, inhibition by antimycin A results in the production of superoxide anion at the Qo site because of the binding of QH2 and the very slow production of semiquinone (DOI: 10.1074/jbc.M111.267898 ).
Lines 438-441: Along these lines, many arguments could be made regarding the biological interactions between SRZ and maize throughout their evolutionary history and if AOX emerged in the plant as a response to fungal infection or if the fungus acquired Aox to assimilate and survive inside the plant tissue.
This proposal is incorrect. The ancestor of plants, fungi, and metazoans contained the AOX gene (doi:10.1093/jxb/err441, doi:10.1016/j.cbd.2006.08.001). This is the reason many non-pathogenic fungi have the AOX gene.
Author Response
Reviewer 2:
Comments and Suggestions for Authors
This work describes the generation of an AOX deletion mutant of the phytopathogenic basidiomycete Sporisorium reilianum and its effect on growth and its involvement in the pathogenic capacity of the fungus. The results show that in the absence of the AOX the serious symptoms of the plant infection are reduced. Expression of the AOX mRNA is higher at the teliospore stage. The manuscript is well written in good English.
We thank the Reviewer for their positive assessment of the paper. We also appreciate the Reviewer’s attention to detail and expertise in providing clearer and more accurate presentation of the biochemistry of the energetics of mitochondrial function.
Next, I have some minor comments:
line 72: The sentence “the AOX is an inefficient proton pump” is inaccurate and ambiguous, because this enzyme is not a proton pump.
We have modified the section as follows: “… aox catalyzes the reduction of oxygen into water but does not function as a proton pump [7]. In this manner, electron flow is limited to complexes I and II, with complex I being the sole contributor to the electrochemical gradient in the mitochondrial intermembrane space.”
Lines 97-100: Additionally, expression analysis of distinct developmental stages of the fungus indicated increased expression in teliospores, corresponding with the use of a less energy-providing respiratory pathway during quiescent stages of the life cycle.
Is there any data indicating that the cytochrome pathway is missing in the teliospore? Has the respiratory capacity of the AOX and the classic pathway been studied in the teliospore?. Even if the AOX is in higher amounts in the teliospore, the enzyme may be in an inactive state. Using an oxygen isotope discrimination technique in plants, it has been shown that AOX has a minor contribution to respiration, even if its capacity is large (https://doi.org/10.3389/fpls.2021.752795). About 10-30%.
We are in the process of adapting respirometry measurements on mitochondria in both haploid sporidia, as well as on teliospores. Until such techniques can be better adapted to these fungi, we are limited to the observations on increases in mRNA levels in teliospores, compared to sporidia and mated cells.
Lines 285-286: Treatment with AMA and/or SHAM resulted in a reduced survival rate in wildtype (WT) strains, although it did not eliminate growth.
I understand that the experimental design in figure 4 measures two things: a) viability after the incubation with the inhibitors and b) growth (observed as an increase in the number of cells after the incubation). So, I agree with the authors that in the presence of AM or SHAM all the strains were capable of growing. However, in the presence of both AM and SHAM, approximately 90% of the WT cells died during the incubation (initial number of cells = 10^5, final number=10^4). Interestingly, this large decrease in cell number only occurred with the WT, but not with the AOX mutants. Fort the mutants, it seems that cells did not grow (maintained the initial number at 10^5), but their viability remained. Is there an explanation?
A second point: According to figure 2, the WT cells have a sufficient amount of AOX. However, the growth of wild-type cells containing the AOX was the same as that of the two AOX deletion mutants in the absence of inhibitors (Figure 4). Should this result indicate that the AOX in the wild-type has no activity or a very small activity during the growth period?
The first thing we would like to point out is that this assay did not take into consideration how much of the inhibitor we were adding to the cultures so that the effect was dramatic (i.e., all cells died if too much inhibitor as used). We tried different concentrations until we identified one that allowed us to quantify the surviving ratio in plate counts using serial dilutions. Further, these results suggest that when growing as haploid sporidia, Aox contribution is less important unless the cells are faced with additional inhibitors of the ETC. Secondly, perhaps it is also worth mentioning that SRZ might have additional components of the ETC, as has been shown in U maydis (notably, external NADH dehydrogenases). The complete absence of Aox in the mutants may be contributing to these additional alternative components to be more active or may make their contribution more important here. Again, it would be very informative to carry out these experiments from a respiratory capacity perspective to pin point what component is driving respiration under the different treatments.
Lines 352-354: These results suggest that Aox-mediated respiration is favored during the teliospore developmental stage, consistent with the lower metabolic demands of a
quiescent life phase
1) I think this proposal is speculative. Is there any data indicating that the cytochrome pathway is missing in the teliospore? Has the respiratory capacity of the AOX and the classic pathway been studied in the teliospore?. Even if the AOX is in higher amounts in the teliospore, the enzyme may be in an inactive state. Using an oxygen isotope discrimination technique in plants, it has been shown that AOX has a minor contribution to respiration, even if its capacity is large. About 10-15%.
We added an additional line to highlight that more experiments are needed to see if higher amounts of Aox means higher respiratory capacity in spores.
2) Despite the presence of an apparently large amount of AOX in the WT (Figure 2), the concentration of mRNA was quite small. In this sense, Nargang reported that an increase in Neurospora AOX mRNA is not reflected in an increase in the protein (https://doi.org/10.1534/g3.119.400522). Following this line of reasoning, it is possible that an increasing the AOX mRNA in the teliospores will not result in a higher content of the protein. The best way to measure this expression is with antibodies against the AOX or the fluorescence of the AOX-gfp.
We would point out that the observation mentioned by the Reviewer about Figure 2 is in the opposite direction as that indicated by Narang, i.e., here amount of Aox (as Aox-Gfp) appears greater than the mRNA levels. But this may be due to the high level of Gfp fluorescence in these constructs.
We do, however, agree with the Reviewer that to truly measure Aox protein levels these will be great follow-up analyses in future studies to better determine whether increased mRNA levels identified here in teliospores translate to increased Aox protein and /or increase Aox enzymatic activity.
Figure 6: If the comparison is with SRZ1 and given the standard deviations or standard errors, I see a significant difference for all the mated samples and the SRZ1 condition. On the other hand, the values of the mated samples are between the values of SRZ1 and SRZ2.
Lines 383-385: Antimycin A does not displace ubiquinone or coenzyme Q. Antimycin A competes with ubiquinone for the Qi site; therefore, it prevents the binding of ubiquinone to this site. As a consequence of the strong binding of antimycin A to the Qi site, the flow of electrons associated with the Q cycle stops. This blockage cannot give rise to a flow of electrons. Certainly, inhibition by antimycin A results in the production of superoxide anion at the Qo site because of the binding of QH2 and the very slow production of semiquinone (DOI: 10.1074/jbc.M111.267898 ).
We have changed the language of this section to reflect the more accurate description characterized by the Reviewer: “AMA is a bacterial secondary metabolite that acts by binding to the Qi site of complex III, thereby competing with coenzyme Q to arrest the movement of electrons.”
Lines 438-441: Along these lines, many arguments could be made regarding the biological interactions between SRZ and maize throughout their evolutionary history and if AOX emerged in the plant as a response to fungal infection or if the fungus acquired Aox to assimilate and survive inside the plant tissue.
This proposal is incorrect. The ancestor of plants, fungi, and metazoans contained the AOX gene (doi:10.1093/jxb/err441, doi:10.1016/j.cbd.2006.08.001). This is the reason many non-pathogenic fungi have the AOX gene.
We have modified this section to better reflect the evolutionary history of Aox indicated by the Reviewer: “…many arguments could be made regarding the biological interactions between SRZ and maize throughout their evolutionary history that may explain the emergence of Aox as an adaptive innovation or as simply the result of divergent evolution.”